# Making Learner Weakness Actionable
# for Learning from Demonstration with Novice Teachers

Yuqing Zhu [1]   Matthew Howard [1]

## Abstract

Learning from demonstration can be an effective way to teach robots task-oriented policies. However, in an interactive setting when demonstrations are limited by time or other budgetary constraints, it is challenging to find those that fix the learner's (remaining) errors. This is especially difficult for novice teachers: they may provide task-valid trajectories, often these fail to meaningfully improve the policy due to their lack of knowledge of learning mechanisms internal to the robot. This paper introduces CLASP (**C**ollaborative **L**earning with **A**nchored **S**tate-space **P**artitions), which summarises the teaching process as a compact map of behavioural regions anchored in the teacher's own demonstrations. The map connects task failure to actionable changes to demonstrations by indicating what is going wrong in an intuitive way. It also enables difficulty-aware training that emphasises regions where learning is failing. Across diverse benchmarks, CLASP improves success by up to 20% over offline and interactive baselines under the same demonstration budget, improves robustness under distribution shift by 14–20%, and preserves behavioural diversity.

## 1. Introduction

Learning from Demonstration (LfD) is a practical route to training robot policies because it uses human examples rather than unsafe trial-and-error. Recent imitation learners, including diffusion policies and transformer-based action generators, can reproduce rich, long-horizon manipulation behaviours when trained on sufficiently diverse demonstrations (Chi et al., 2025; Zhao et al., 2023). However, these

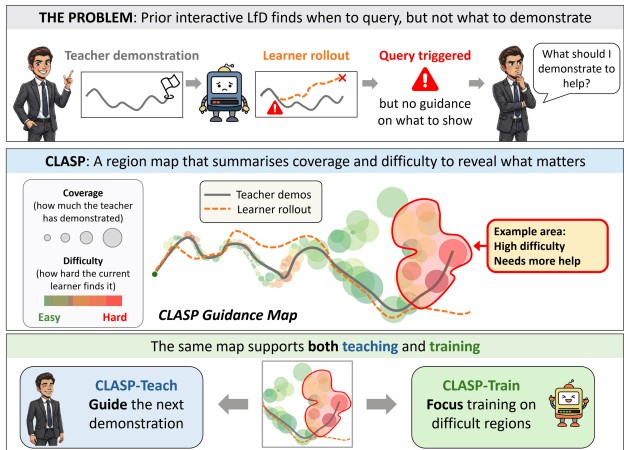

*Figure 1.* Traditional interactive LfD decides when to query using learner-side signals, but does not tell novices what to demonstrate. CLASP converts failures into teacher-facing guidance.

gains remain closely tied to data scale: large cross-robot corpora and generalist policies demonstrate impressive breadth, but also highlight the cost of collecting and curating demonstrations (O'Neill et al., 2024; Mees et al., 2024). In realistic settings, demonstrations are often unevenly distributed, with biased or redundant coverage that leaves critical behaviours under-sampled and policies brittle under distribution shift (Aliasghari et al., 2024; Hou et al., 2023).

This has pushed recent work toward more *selective* supervision. Interactive imitation learning lets learners request additional demonstrations online, often around failures or high uncertainty, so new data is gathered near the policy's error-prone regions (Luijkx et al., 2025; Cai et al., 2025; Hou et al., 2025b). Curriculum-based methods use demonstrations to structure practice or query selection, typically progressing from easier cases to harder ones (Hou et al., 2025a; Tao et al., 2024; Bauza et al., 2025). Methods also re-weigh, mix, or otherwise curate heterogeneous datasets to improve robustness when demonstrations vary in quality, strategy, or source (Hoang et al., 2024; Hejna et al., 2025). Despite this progress, a practical gap remains: supervision is usually driven by *learner-centric* signals (*e.g.,* uncertainty, intervention criteria, training schedules, mixture weights) rather than a *teacher-interpretable* account of which behaviours the learner fails to imitate. Under time or budget

[1]Department of Engineering, King's College London, London, UK. Correspondence to: Yuqing Zhu <yuqing.zhu@kcl.ac.uk>, Matthew Howard <matthew.j.howard@kcl.ac.uk>.

*Proceedings of the 43rd International Conference on Machine Learning*, Seoul, South Korea. PMLR 306, 2026. Copyright 2026 by the author(s).

limits, the learner may localise errors, but the teacher still lacks actionable guidance on what to demonstrate next.

This gap is most acute when robots are taught by *novice teachers*, which is common in many deployments (Sakr et al., 2025). Novices can usually judge whether a demonstration succeeds, but success does not imply that it is informative for the current learner. Informativeness is policy-dependent: demonstrations matter most when they cover behaviours the learner still fails to imitate, such as corner cases and corrective actions (Ross et al., 2011; Hoque et al., 2025). However, without prompts, novices often default to learner-agnostic coverage (*e.g.,* allocating demonstrations across plausible strategies rather than targeting the learner's failure modes), so errors persist even as data accumulates. This aligns with evidence that imitation performance is sensitive to demonstration diversity and quality (Belkhale et al., 2023), and that user interfaces and incentives can bias which demonstrations novices provide (Mirchandani et al., 2025). From a Machine Teaching perspective (Zhu, 2015), the missing piece is a teacher-interpretable representation that reveals where additional demonstrations would most reduce imitation error.

To address this, we introduce **CLASP**, a guidance framework that helps teachers resolve imitation failures under a fixed demonstration budget (Figure 1). CLASP builds a compact map coupling *coverage* with *difficulty*, with regions defined jointly by teacher demonstrations and the learner's on-policy behaviour. The map exposes recurring failure patterns and highlights where additional demonstrations are most likely to improve learning. Across simulation and real-robot benchmarks, CLASP increases success by up to 20% over interactive and offline baselines under the same demonstration budget, and improves robust success under distribution shift by 14–20% while maintaining high behavioural diversity. The contributions are: (i) a human-anchored, learner-aware map that shows failure patterns in terms of observable robot behaviour, rather than opaque internal learning signals, (ii) a teaching policy that directs novice effort toward high-impact regions, and (iii) a coupled teaching and training mechanism where the same map guides both which demonstrations to request and which errors to prioritise during training, all within a fixed demonstration budget.

## 2. Related Work

**Selective supervision in interactive LfD.** A natural way to reduce human effort in LfD is to be selective about when supervision is requested, using learner-side signals to gate queries. ASkDAgger, for example, asks for supervision only when the learner is unsure, so queries tend to yield corrective labels rather than duplicating what is already covered (Luijkx et al., 2025). AIM pursues a similar goal, but

learns an intervention mechanism to decide when a human should step in, instead of relying on a hand-set uncertainty rule (Cai et al., 2025). Building on dataset aggregation and confidence-based autonomy (Ross et al., 2011; Chernova & Veloso, 2009), this direction has expanded to methods that request supervision when uncertainty, novelty, or risk is high (Zhang & Cho, 2016; Menda et al., 2019; Hoque et al., 2025), as well as human-gated approaches where the teacher intervenes at the right moments (Kelly et al., 2019; Mandlekar et al., 2020). Subsequent work amplifies interventions into additional corrective data (Hoque et al., 2024), and recent work studies calibrated help-seeking in language-instructed settings (Ren et al., 2023). Across these approaches, the common strength is efficient supervision through query timing; however, a recurring limitation, especially with novice teachers, is communicative: triggers can decide *when* to ask, yet still leave the teacher without a clear behavioural account of *what* to demonstrate next.

**Curricula and teacher guidance.** Many methods increase demonstration informativeness by shaping what people demonstrate. Curriculum-based approaches stage requests from easier cases to harder ones, so limited supervision is spent on situations expected to drive learning progress (Hou et al., 2025a; Tao et al., 2024; Bauza et al., 2025). Other systems support novice teachers more directly, for example, by leading users toward likely failure cases (Sakr et al., 2025) or enabling real-time corrections during execution (*e.g.,* LILAC) (Cui et al., 2023). Evidence from large-scale collection further shows why teacher-facing design matters: interfaces and incentives shape the strategies that novices default to, affecting both quality and diversity (Li et al., 2025; Mirchandani et al., 2025). CLASP complements design issues by grounding guidance in the teacher's demonstrated behaviours and highlighting regions where the learner still struggles, refining natural teaching habits without requiring wholesale changes to teaching strategy.

**Offline curation and machine teaching.** Mixed-quality demonstrations are common, and many LfD methods improve robustness via re-weighting or mixing imperfect data (Hoang et al., 2024; Hejna et al., 2025; Belkhale et al., 2023). However, for novice teachers, learner-side curation does not resolve the issue of deciding what to demonstrate next. Machine teaching makes this gap explicit by treating data selection and the teaching protocol as part of the learning problem (Zhu, 2015), and recent work leverages this idea to upskill users by providing training guidance (Sun et al., 2025; Zhu et al., 2024). While promising, these methods often rely on simplified settings or pre-defined expert guidance. CLASP is inspired by this perspective and targets more general LfD settings.

*Figure 2.* **CLASP overview.** At each round $k$, the learner trains on the current dataset $\mathcal{D}_k$ and rolls out to collect on-policy states. CLASP then constructs a region map $\mathcal{C}_k$ coupling coverage with difficulty, and uses it for (i) teacher guidance to collect new demonstrations (CLASP-Teach) and (ii) difficulty-aware reweighting during training (CLASP-Train).

## 3. Problem Formulation

In this paper, LfD is considered in a sequential decision-making setting with observation space $\mathcal{X}$ and action space $\mathcal{A}$. At each time step, the learner receives an observation $x \in \mathcal{X}$ (including task context such as a goal specification) and outputs an action $a \in \mathcal{A}$ according to a policy $\pi_\theta : \mathcal{X} \to \mathcal{A}$.

Learning is supervised by a human teacher through demonstrations. Each demonstration is a labelled trajectory of fixed length $T$

$$\tau = (x_1, a_1, \ldots, x_T, a_T), \qquad |\tau| = T,$$

and the demonstration dataset is $D = \{\tau^{(i)}\}_{i=1}^N$. The learning algorithm $A$ is assumed fixed and trains a policy from demonstrations,

$$\pi_\theta = A(D).$$

Because human behaviour can be multi-strategy, multiple successful ways of acting may be valid. This variability is captured through an unknown distribution $P_h$ over observation–action pairs $(x, a)$, representing the behaviours the learner should imitate. To measure how well a learnt policy matches this target behaviour, a per-step loss $\ell(a, \pi_\theta(x))$ is used and the resulting *teaching risk* is

$$R_T(\pi_\theta) = \mathbb{E}_{(x,a)\sim P_h}\left[\ell\big(a, \pi_\theta(x)\big)\right].$$

Demonstrations are costly to obtain, so teaching effort is also accounted for. Effort is measured by the total number of data points,

$$E_T(D) = \sum_{\tau \in D} |\tau|.$$

The goal is to select a dataset that induces a low-risk policy while respecting the demonstration budget:

$$D^\star \in \arg\min_D \; R_T\big(A(D)\big) \quad \text{s.t.} \quad E_T(D) \le B. \quad (1)$$

Equivalently, the Lagrangian relaxation

$$\min_D \; R_T\big(A(D)\big) + \lambda E_T(D),$$

may be used to trade off imitation performance against demonstration effort.

## 4. CLASP: Region-Level Machine Teaching

The difficulty in optimising (1) is structural: the teacher's decision variable is the dataset itself, while the quantity of interest depends on the policy obtained after training on that dataset. This bi-level dependence makes optimal teaching difficult in realistic LfD settings, where demonstrations are high-dimensional, multimodal, and expensive to acquire. CLASP addresses this by introducing a region-level interface between the teacher and the learner, as shown in Figure 2. Each region summarises two signals: coverage (what has been demonstrated) and difficulty (where the current learner struggles). This shared "map" supports two operations in each round: **CLASP-Teach** uses it to decide where new demonstrations are most valuable, and **CLASP-Train** uses it to focus learning on what remains hard (see Appendix A.2).

Formally, let $D_k$ denote the demonstration data at teaching round $k$ and $\pi_{\theta_k} = A(D_k)$ the current policy. For any demonstrated step $(x_t, a_t)$ from any trajectory $\tau \in D_k$, a local difficulty signal is defined using the per-step loss

$$d_k(x_t, a_t) \;=\; \ell\big(a_t, \pi_{\theta_k}(x_t)\big). \quad (2)$$

Large values of $d_k$ indicate local mismatch between the learner and the teacher, while small values indicate local agreement. Difficulty is computed by querying the current learner on *teacher-provided observations*, yielding a teacher-aligned notion of what has not yet been mastered. This signal is aggregated over neighbourhoods to identify what the learner currently finds difficult.

Neighbourhoods are defined in the normalised embedding

$$\phi_k(x) \;=\; \frac{x - \mu_k}{\sigma_k + \epsilon},$$

where $\mu_k$ and $\sigma_k$ are the per-dimension mean and standard deviation computed over the explored set $U_k$ (defined below), and $\epsilon > 0$ is a small constant. Locality is measured by Euclidean distance in the embedded space.

Region construction uses both demonstrated observations and those induced by the current policy. Let

$$X_H^{(k)} = \big\{\, x_t \in \mathcal{X} \;:\; \tau \in D_k,\, t \le |\tau| \,\big\}$$

denote the observations appearing in demonstrations, treated as a multiset. Let $X_M^{(k)}$ denote the observations encountered in rollouts of $\pi_{\theta_k}$ on sampled tasks. These are combined into an *explored set*

$$U_k \;=\; X_H^{(k)} \cup X_M^{(k)}, \tag{3}$$

which captures both (i) what has been demonstrated and (ii) the states the learner currently visits when acting. The $(\mu_k, \sigma_k)$ in $\phi_k$ are computed over $U_k$, so locality adapts to the scale of both demonstration data and current rollouts.

## 4.1. Teaching representation

Given $U_k$ and $d_k$, a region-level view of the current teacher–learner interaction is constructed. Each region is specified by (i) a neighbourhood radius that adapts to local coverage and (ii) an aggregated difficulty score. Regions are human-anchored (built around demonstrated observations) while their geometry is learner-aware (scaled using $U_k$), coupling coverage and error in shared local coordinates. This representation is used by both **CLASP-Teach** and **CLASP-Train**.

**Region geometry.** A region should represent a local portion of behaviour, but locality in LfD is highly non-uniform: demonstrations and rollouts often cluster around some modes while leaving other modes sparse. To this end, a locally adaptive radius based on $K$-nearest neighbours (KNN) is used as a non-parametric estimate of local scale.

Specifically, let $x \in X_H^{(k)}$ be an anchor observation and let $\mathcal{N}_k(x)$ be the set of its $K_{\mathrm{nn}}$ nearest neighbours in $U_k$. The radius is set as the $q$-quantile of the corresponding neighbour distances,

$$r_k(x) \;=\; \mathrm{Q}_q\Big(\big\{\|\phi_k(x) - \phi_k(x')\|_2 : x' \in \mathcal{N}_k(x)\big\}\Big). \tag{4}$$

This retains adaptivity while being robust to outliers. The region is then defined as the induced ball over the explored set,

$$\mathcal{B}_k(x) \;=\; \big\{\, x' \in U_k : \|\phi_k(x') - \phi_k(x)\|_2 \le r_k(x) \,\big\}. \tag{5}$$

Using $U_k$ in the KNN computation ensures that region geometry reflects both what has been demonstrated and what the current policy visits, while anchoring regions at $X_H^{(k)}$ keeps the representation tied to teacher-provided behaviour.

**Region difficulty.** A region score should reflect whether the learner fails anywhere within the region, but naive choices have drawbacks. An average can dilute rare-but-important mistakes, especially in multi-strategy data where easy modes dominate, while a hard maximum is sensitive to a single noisy label or a transient prediction spike. A *soft maximum* is therefore used to emphasise difficult points while remaining stable.

To avoid ambiguity from repeated observations, region coverage is defined at the level of demonstration steps. Let

$$\mathcal{I}_k = \big\{\, (\tau, t) \;:\; \tau \in D_k,\, 1 \le t \le |\tau| \,\big\}$$

index demonstrated steps in round $k$, where $(\tau, t)$ refers to the $t$-th step of trajectory $\tau$. The demonstrated pairs covered by $\mathcal{B}_k(x)$ are defined as

$$\mathrm{cov}_k(x) = \big\{\, (x_t, a_t) \;:\; (\tau, t) \in \mathcal{I}_k,\, x_t \in \mathcal{B}_k(x) \,\big\}. \tag{6}$$

By construction, $\mathrm{cov}_k(x) \ne \emptyset$. The region difficulty is then defined using a size-normalised soft maximum,

$$\tilde{d}_k(x) \;=\; \log\left(\frac{1}{|\mathrm{cov}_k(x)|} \sum_{(x,a) \in \mathrm{cov}_k(x)} \exp\big(d_k(x, a)\big)\right). \tag{7}$$

The region descriptor is written as

$$c_k(x) \;=\; \big(x, r_k(x), \tilde{d}_k(x)\big),$$

and the collection of anchored region descriptors forms the region map

$$\mathcal{C}_k \;=\; \big\{\, c_k(x) : x \in X_H^{(k)} \,\big\}. \tag{8}$$

## 4.2. CLASP-Teach: Region-based teaching policy

CLASP-Teach converts the region map $\mathcal{C}_k$ into guidance that a human teacher can act on. Since $\mathcal{C}_k$ contains many overlapping regions, showing all of them would be redundant and difficult to interpret. CLASP-Teach therefore selects a set of representative regions that are (i) difficult for the current learner and (ii) complementary in the demonstrated *steps* they cover.

**A weighted coverage objective.** Each anchor $x$ defines a region $\mathcal{B}_k(x)$ with aggregated difficulty $\tilde{d}_k(x)$. Demonstrated steps $\mathcal{I}_k$ are treated as the items that selected regions should cover. For a set of uncovered demonstrated steps $\bar{\mathcal{I}} \subseteq \mathcal{I}_k$, the marginal coverage of $x$ is defined as

$$\Delta_k(x; \bar{\mathcal{I}}) = \big|\{\, (\tau, t) \in \bar{\mathcal{I}} : x_t \in \mathcal{B}_k(x) \,\}\big|. \tag{9}$$

Each candidate anchor is scored by the teaching gain

$$G_k(x; \bar{\mathcal{I}}) = \tilde{d}_k(x)\, \Delta_k(x; \bar{\mathcal{I}}). \qquad (10)$$

This objective favours regions that are both difficult (high $\tilde{d}_k$) and non-redundant (large marginal coverage over uncovered steps).

**Selection of regions to present.** Let $L_k$ denote the number of regions presented in round $k$. Initialising $\mathcal{Q}_k = \emptyset$ and $\bar{\mathcal{I}} = \mathcal{I}_k$, for $\ell = 1, \dots, L_k$, an unchosen anchor

$$x^\star \in \arg \max_{x \in X_H^{(k)}:\, c_k(x) \notin \mathcal{Q}_k} G_k(x; \bar{\mathcal{I}}), \qquad (11)$$

is selected. After adding $c_k(x^\star)$ to $\mathcal{Q}_k$, the uncovered set is updated by removing demonstrated steps covered by $\mathcal{B}_k(x^\star)$:

$$\bar{\mathcal{I}} = \bar{\mathcal{I}} \setminus \big\{ (\tau, t) \in \bar{\mathcal{I}} : x_t \in \mathcal{B}_k(x^\star) \big\}.$$

The procedure is stopped if $\bar{\mathcal{I}}$ becomes empty. The resulting $\mathcal{Q}_k \subseteq \mathcal{C}_k$ is a compact set of difficult, diverse regions.

In the interface, $\mathcal{Q}_k$ is visualised as highlighted neighbourhoods around their anchors (with difficulty and radius), indicating where additional demonstrations are likely to help.

### 4.3. CLASP-Train: Difficulty-aware training

Beyond guiding where to collect additional demonstrations, the region map $\mathcal{C}_k$ provides an intuitive way to emphasise demonstrated states that the current learner still finds difficult. In round $k$, each demonstrated step is weighted according to the difficulty of its associated region.

Because regions are anchored at demonstrated observations, each demonstrated step $x_t$ is associated with an anchor

$$\alpha_k(x_t) \in \arg \min_{x \in X_H^{(k)}} \|\phi_k(x_t) - \phi_k(x)\|_2.$$

If multiple anchors are at the same minimum distance, one is selected. Step $x_t$ then inherits the difficulty of its assigned anchor, *i.e.*, $\tilde{d}_k(\alpha_k(x_t))$. Let

$$\bar{d}_k = \frac{1}{|X_H^{(k)}|} \sum_{x' \in X_H^{(k)}} \tilde{d}_k(x')$$

denote the average region difficulty over the demonstrated observations. The difficulty-aware weight is

$$w_k(x_t) = 1 + \frac{\tilde{d}_k(\alpha_k(x_t))}{\bar{d}_k + \varepsilon}, \qquad (12)$$

where $\varepsilon > 0$ is a small constant for numerical stability.

The learner is then updated by minimising the weighted imitation objective on $D_k$,

$$\mathcal{L}_k(\theta) = \frac{1}{\sum_{\tau \in D_k} |\tau|} \sum_{\tau \in D_k} \sum_{t=1}^{|\tau|} w_k(x_t)\, \ell\big(a_t, \pi_\theta(x_t)\big), \qquad (13)$$

implemented by weighting per-sample losses within each minibatch by $w_k(x_t)$.

Together, CLASP-Teach and CLASP-Train use $\mathcal{C}_k$ to couple data collection and optimisation: regions identified as difficult are highlighted for further demonstration and simultaneously receive greater emphasis during training.

## 5. Experimental Setting

**Baselines.** In this section, CLASP is compared against representative strategies for allocating a fixed demonstration budget, including learner-gated querying, curriculum scheduling, and offline curation. For selective querying, an uncertainty-gated baseline in the style of ASkDAgger is used (Luijkx et al., 2025). For curricula, an easy-to-hard baseline is used, where demonstration collection is staged over predefined task-specific difficulty bins (Hou et al., 2025a; Tao et al., 2024; Bauza et al., 2025). For offline curation, two training-only heuristics are evaluated on a fixed dataset: (i) trajectory-quality weighting, which assigns each trajectory a scalar weight based on outcome and efficiency, and (ii) cluster-balanced re-weighting, which clusters demonstrations into behavioural groups and weights samples by inverse cluster frequency (Belkhale et al., 2023; Hoang et al., 2024; Hejna et al., 2025). Finally, a no-guidance condition is included where novice teachers demonstrate using their own judgement under the same budget and interface; this dataset is kept unchanged in the offline curation baselines to isolate training-only effects. Additional details are provided in Appendix A.3.

**Benchmarks.** Evaluations are conducted in a MuJoCo manipulation suite drawn from D3IL (Jia et al., 2024): AVOIDING, PUSHING, PICK-AND-PLACE, and ALIGNMENT (as shown in Figure 3). These tasks share the same state/action interface but differ in contact dynamics and precision demands. In addition, results are also reported for training the Sawyer robot on AVOIDING and STIRRING (see Appendix C).

**Experimental protocol.** Budgeted teaching is studied in a round-based protocol with novice participants (no prior robotics teleoperation experience). In simulation, four between-subject groups (5 participants each) are used: **CLASP**, **ASkDAgger**-style querying, **Curriculum**, and **No-guidance**. For hardware evaluation, two groups (5 participants each) are used: **CLASP** and **No-guidance**. For each task, five rounds are run with cumulative budgets $B \in \{4, 8, 12, 16, 20\}$ trajectories. At each round, a diffusion policy is trained on the current dataset with a fixed compute budget, method-specific guidance (or no guidance) is produced, 4 new trajectories are collected, and the policy is retrained on the updated dataset. Teachers receive no performance feedback between rounds beyond the guidance. Task

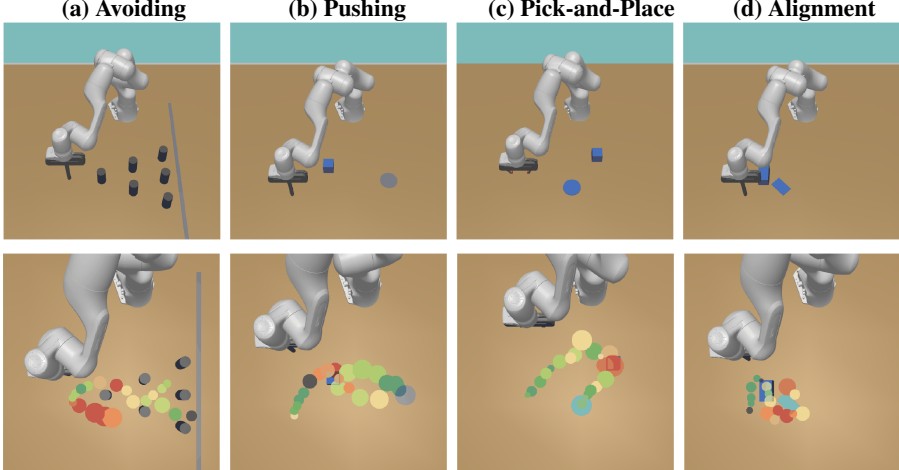

**(a) Avoiding**  **(b) Pushing**  **(c) Pick-and-Place**  **(d) Alignment**

*Figure 3.* Benchmarks and CLASP region visualisation. **Top:** the four simulated manipulation tasks. **Bottom:** an example CLASP region map over a demonstrated trajectory, where circle *colour* indicates region difficulty and circle *size* reflects the coverage measure.

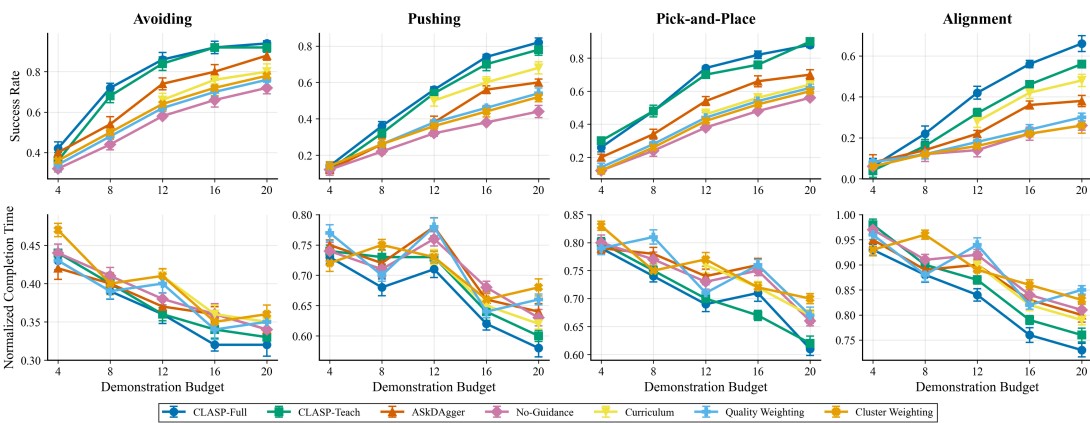

*Figure 4.* Budgeted imitation performance in simulation across the four tasks (shown are mean±s.d.). **Top:** Success rate. **Bottom:** Normalised completion time on successful episodes.

horizons are fixed (AVOIDING 400, PUSHING 400, PICK-AND-PLACE 600, ALIGNMENT 1000). For query-based baselines, states encountered in rollouts are scored, the top-4 are selected, the environment is reset to each snapshot, and the teacher provides a continuation trajectory. The curriculum baseline allocates early rounds to easier instances defined by difficulty bins, then collects from the default distribution in later rounds. **CLASP-Full** and **CLASP-Teach** use the same demonstrations collected under CLASP guidance, differing only in that CLASP-Teach trains without difficulty-aware reweighing. Full implementation details are provided in Appendix A.

**Evaluation metrics.** Each policy is evaluated on a fixed set of test initialisations, and success rate and normalised completion time on successful trials are reported. Completion time is defined as the number of environment steps until success, normalised by the task horizon. To assess robustness, success under distribution shift is also reported by perturbing initial states by $\pm 10\%$ of the default evaluation

initialisation range. Behavioural diversity on AVOIDING and PUSHING is also measured: for each policy, 50 rollouts are completed and assigned to one of two known strategy labels using a fixed geometric rule. Let $n_1$ and $n_2$ denote the rollouts of strategy 1 and strategy 2, and the balance score $r = \min(n_1, n_2)/\max(n_1, n_2)$ is reported.

## 6. Results

**Budgeted performance.** Figure 4 (top) reports success rate as a function of the cumulative demonstration budget $B$ in the simulation tasks. Across all four tasks, CLASP achieves higher success at the same budget and improves faster once guided collection begins. The gap is most pronounced on AVOIDING at moderate budgets, where the learning signal is scarce and demonstration placement matters most. For example, at $B = 12$ CLASP reaches 0.86 on AVOIDING, outperforming ASkDAgger and no-guidance. On the more challenging ALIGNMENT task, CLASP attains the highest

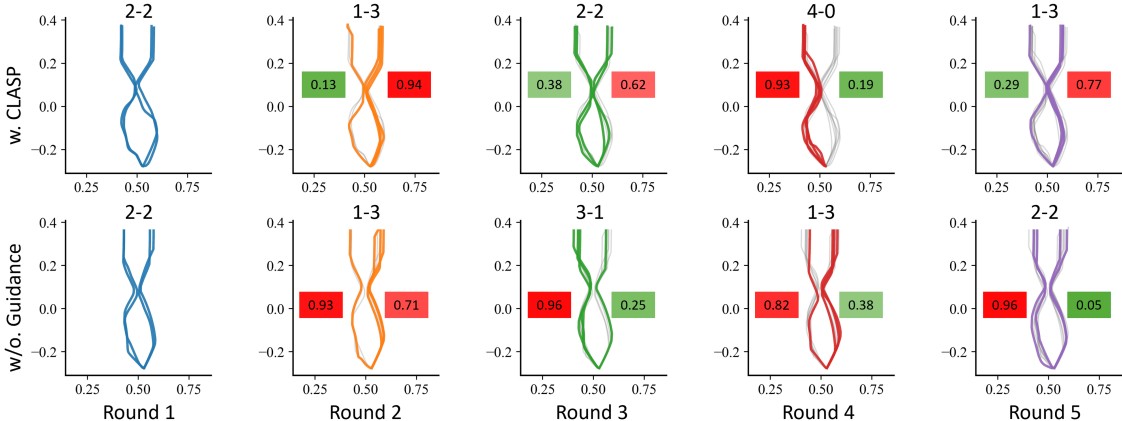

*Figure 5.* Teacher demonstrations over rounds on AVOIDING with CLASP (top) and without guidance (bottom). Each subplot shows trajectories for the two strategy regions, with the header indicating Strategy 1 count – Strategy 2 count. The red/green boxes report the average difficulty of the top-10 CLASP guidance regions, summarizing how difficult each strategy is for the learner.

final success at $B = 20$, while offline-only curation yields smaller gains, consistent with the limitation that reweighing cannot change which trajectories are collected.

Comparisons between CLASP-Full and CLASP-Teach clarify the source of improvement. The success rate of CLASP-Teach matches that of CLASP-Full at larger budgets (*e.g.,* AVOIDING 0.92 vs. 0.94 at $B = 20$), while Full is consistently better in both success rate and completion time on harder cases where targeting and training efficiency jointly matter (*e.g.,* ALIGNMENT 0.66 vs. 0.56 at $B = 20$). Overall, these results demonstrate that CLASP converts policy-relative failure modes into actionable demonstration targets, achieving higher success earlier under the same fixed teaching and training budget.

**Completion time.** Figure 4 (bottom) reports normalised completion time on successful episodes (mean±s.d.). Unlike success rate, completion time reflects execution efficiency and is noisier because the set of successful rollouts changes with budget.

Despite this, normalised completion time generally decreases for CLASP as the budget increases, with the clearest gains on harder, contact-sensitive tasks. On ALIGNMENT, CLASP-Full achieves faster successful executions at larger budgets than the interactive baselines, indicating fewer corrective adjustments near contact. On PUSHING, CLASP-Full also demonstrates shorter successful rollouts, suggesting smoother contact transitions once a successful strategy is learned. For easier tasks such as AVOIDING, completion times converge across methods as success saturates (*e.g.,* 0.32 for CLASP-Full vs. 0.34 for no-guidance at $B = 20$), and remaining differences largely reflect alternative successful styles. Overall, these trends are consistent with the success curves: CLASP's higher success is not achieved by overly cautious policies. Conditioning on suc-

*Table 1.* Robust success rate under distribution shift at the final budget $B = 20$. Best in **bold**.

| Method | Avoiding | Pushing | P&P | Alignment |
|---|---|---|---|---|
| **CLASP-Full** | **0.88** | **0.72** | **0.78** | **0.54** |
| CLASP-Teach | 0.86 | 0.68 | 0.74 | 0.42 |
| ASKDAgger | 0.82 | 0.50 | 0.58 | 0.22 |
| No Guidance | 0.62 | 0.34 | 0.44 | 0.16 |
| Curriculum | 0.74 | 0.58 | 0.54 | 0.36 |
| Quality Weighting | 0.68 | 0.44 | 0.50 | 0.20 |
| Cluster Weighting | 0.70 | 0.42 | 0.48 | 0.18 |

cessful episodes, CLASP achieves completion times that are comparable to or lower than other baselines, suggesting the learnt behaviours better match the teacher's execution efficiency rather than relying on slow corrective motions.

**Robustness under distribution shift.** To test robustness (Table 1), the initial state is perturbed and rollouts are evaluated (AVOIDING: end-effector start; PUSHING/PICK-AND-PLACE: cube start position; ALIGNMENT: cube position and orientation). Under this limited-budget setting ($B \leq 20$), all methods degrade relative to in-distribution performance, but CLASP remains consistently more robust across tasks. At $B = 20$ under shift, CLASP-Full achieves the highest robust success across all four tasks. In contrast, ASkDAgger and no-guidance degrade substantially more, especially on the contact-sensitive ALIGNMENT task. Among non-CLASP baselines, curriculum is typically strongest under shift (*e.g.,* 0.74 on AVOIDING and 0.58 on PUSHING) but still lags CLASP, consistent with CLASP focusing limited demonstrations on policy-relevant failures rather than improving only nominal coverage.

**Behavioural diversity.** Behavioural diversity on AVOIDING and PUSHING is measured using the strategy balance score $r$, where $r = 1$ indicates perfectly balanced use of the two

*Table 2.* **Real-robot performance.** Each entry shows **success rate / normalised completion time** (mean±std).

| Task | Method | B=4 | B=8 | B=12 | B=16 | B=20 |
|---|---|---|---|---|---|---|
| Avoiding | CLASP-Full | 0.32 / 0.48±0.26 | 0.58 / 0.43±0.22 | 0.74 / 0.40±0.19 | 0.88 / 0.36±0.16 | 0.92 / 0.35±0.17 |
| | No-Guidance | 0.38 / 0.51±0.28 | 0.46 / 0.47±0.25 | 0.54 / 0.41±0.25 | 0.62 / 0.40±0.21 | 0.76 / 0.38±0.14 |
| Stirring | CLASP-Full | 0.18 / 0.85±0.32 | 0.34 / 0.78±0.29 | 0.50 / 0.71±0.15 | 0.64 / 0.67±0.19 | 0.76 / 0.62±0.21 |
| | No-Guidance | 0.14 / 0.83±0.28 | 0.26 / 0.90±0.34 | 0.36 / 0.86±0.21 | 0.58 / 0.74±0.23 | 0.68 / 0.68±0.17 |

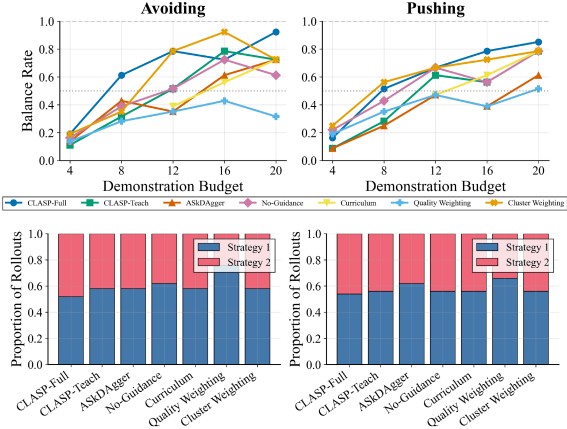

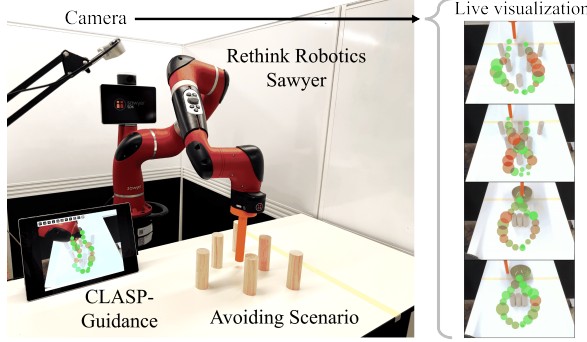

*Figure 7.* The real-robot setting and CLASP guidance. See more details in Appendix B

*Figure 6.* Behavioural diversity on AVOIDING and PUSHING. **Top:** strategy balance score $r$ versus demonstration budget $B$ (higher is more balanced). **Bottom:** final strategy usage proportions at $B = 20$ across methods.

strategies and smaller values indicate mode collapse (Figure 6). CLASP reaches high balance with fewer rounds and achieves the strongest final diversity. On both tasks, CLASP-Full increases $r$ substantially with budget and achieves the highest final balance, while ASkDAgger and No-Guidance remain more skewed toward a single strategy. Among baselines, Cluster Weighting and Curriculum can sometimes improve balance, but gains are less consistent across rounds. Diversity is interpreted jointly with success: in this setting, CLASP's higher balance accompanies higher success, suggesting the guidance supports multiple successful strategies rather than collapsing to the dominant mode.

**Teacher behaviour over rounds.** Figure 5 illustrates that CLASP helps novice teachers adapt their demonstration allocation across rounds: as guidance reveals where the learner struggles, teachers increasingly provide demonstrations in the weaker strategy region, reducing mode bias and improving balance between strategies. In contrast, without guidance, teachers follow their own sampling habits, resulting in a more arbitrary distribution of demonstrations and, in some cases, an even stronger collapse toward the dominant strategy. These results suggest that CLASP's gains come from converting additional demonstrations into *targeted corrective feedback* rather than accumulating redundancy.

**Real-robot results.** Table 2 summarises real-robot success rate and normalised completion time. In hardware, we de-

ploy CLASP by overlaying the guidance directly onto the robot's camera view (Figure 7), providing an intuitive interface that transfers without task-specific tuning. Consistent with simulation, this guidance yields reliable performance improvements over No-Guidance as the demonstration budget increases, with clear gains on both tasks.

# 7. Conclusion

This paper presented CLASP, a guidance framework for data-efficient LfD under small trajectory budgets, where performance is often constrained by which demonstrations are collected. Region-level difficulty summarises weaknesses and converts them into actionable requests, enabling novice teachers to prioritise failure regions while avoiding redundant data. Across manipulation tasks, success is consistently improved over no-guidance and interactive/offline baselines under the same budget. CLASP-Full achieves the strongest overall performance, while CLASP-Teach captures most of the gains without difficulty-aware reweighting. Rollout efficiency on successful trials is improved, and higher success is maintained under moderate initial-state perturbations.

**Limitations and future work.** CLASP may require adaptation for substantially longer-horizon tasks or more complex observation spaces. Also, diversity evaluation uses task-specific strategy labels. Extending the framework to automatically discover behavioural modes and support richer curricula is an important next step. Finally, while novice teachers are studied in both simulation and hardware, scalability to more heterogeneous users, robots and environments remains to be examined, including interface designs for higher-dimensional guidance and longer training horizons.

## Acknowledgments

This work was supported by King's College London and the China Scholarship Council.

## Impact Statement

Robot learning is trending toward broader task coverage by scaling data collection through demonstrations, interventions, and mixed offline/online training. In this setting, the limiting factor is often not model capacity but supervision quality under practical constraints: real-world trajectories are expensive to obtain, teachers (especially non-experts) naturally repeat comfortable behaviours, and failures can be rare or concentrated in specific parts of the state space. As a result, additional data may have diminishing returns unless it is targeted toward the learner's current weaknesses. Guidance systems that translate "what the policy struggles with" into "what a person should demonstrate next" therefore play an enabling role in making data collection more efficient, more repeatable across teachers, and easier to integrate into iterative robotics workflows.

The primary societal benefit of such guidance is reducing the human time and expertise required to train reliable manipulation policies, which can improve accessibility of robot learning beyond expert laboratories. Potential downsides are limited but worth noting: guidance may bias collected data toward a teacher's preferred strategies or overlook low-frequency edge cases if followed uncritically. These risks are mitigated in typical LfD pipelines by routine evaluation on held-out initial states and by aggregating demonstrations across rounds or users, and the method itself does not introduce new sensing, autonomy, or deployment capabilities. Overall, we expect low risk and positive impact when used as decision support during data collection and paired with standard validation practices.

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

## A. Experimental Details

### A.1. Policy & Training Hyperparameters

This section summarises the diffusion policy architecture, training hyperparameters, and task-specific observation/action interfaces used across all experiments.

**Diffusion policy.** We use a state-based Transformer diffusion policy with an encoder–decoder backbone. The policy takes the current state observation (task-dependent dimension) and predicts a sequence of 20 future actions in

delta-position (velocity) form. Diffusion uses 100 denoising steps with a cosine beta schedule (offset $s=0.008$) and $\epsilon$-prediction. The backbone consists of a 1-layer Transformer encoder and a 2-layer Transformer decoder with embedding dimension 256, 4 attention heads, and dropout 0.1 (attention/residual). Time conditioning uses sinusoidal timestep embeddings followed by a 2-layer MLP with Mish activations. Training uses AdamW with L2 loss, and inference uses EMA weights.

**Hyperparameters.** Table 3 summarises the core diffusion model and training hyperparameters. We keep compute per round fixed by training each round for 100 epochs with identical optimisation settings.

**Tasks, environments, and preprocessing.** Table 4 lists the task-specific observation definition and dimensionality, along with the corresponding action interface. All tasks use delta-position (velocity) actions. For pick-and-place, the policy also outputs a gripper command that is stabilised at execution time to avoid high-frequency switching. Specifically, we apply a hysteresis (debouncing) rule: the gripper state is binary (`open`/`close`) and is updated only when the predicted gripper value crosses two asymmetric thresholds, switching to `close` if $g < \tau_{\text{close}}$ and to `open` if $g > \tau_{\text{open}}$, with $\tau_{\text{close}} < \tau_{\text{open}}$ (we use $\tau_{\text{close}}=0.05$ and $\tau_{\text{open}}=0.07$). If $g$ lies between thresholds, the previous gripper state is held. Table 5 reports episode horizons (used for normalising completion time) and workspace bounds per task. We apply z-score normalisation (zero mean, unit variance) to both states and actions, with statistics computed on the training set. Trajectories are segmented into overlapping chunks of length 20, producing training pairs of the form $(o_t, a_{t:t+19})$. Action sequences are computed as position deltas between consecutive states; for pick-and-place, XYZ actions are deltas while the gripper channel is treated as an absolute value before hysteresis filtering during execution. Note, angle wrapping was not encountered in our pick-and-place trajectories; we therefore use the raw angle value.

## A.2. CLASP Implementation Details

This section summarises the implementation-level details of CLASP needed for reproducibility but omitted from the main text, including (i) how the explored set is instantiated each round, (ii) how anchor neighbourhoods and radii are computed in practice, and (iii) the end-to-end pipeline used to generate teacher guidance and apply difficulty-aware weighting during training (Algorithm 1).

**Per-round data used for guidance.** Guidance is recomputed *once per round* after updating the learner. We form the explored set by combining (i) all demonstrated states so far and (ii) the most recent on-policy rollout states generated by the current learner in the same environment setting as evaluation. Unless otherwise stated, we retain all demon-

---

**Algorithm 1** CLASP implementation

---

**Require:** Demonstrations $D_0$; learner update $A(\cdot)$; rounds $K$; `Knn`=30; $q=0.95$; clip $w_{\max}$
**Ensure:** Policies $\{\pi_{\theta_k}\}_{k=0}^{K}$ and displayed guidance regions $\{Q_k\}_{k=0}^{K-1}$
1: Train initial learner $\pi_{\theta_0} \leftarrow A(D_0)$
2: **for** $k = 0$ **to** $K - 1$ **do**
3:      Roll out $\pi_{\theta_k}$ to collect on-policy states $X_M^{(k)}$
4:      Extract demonstrated states $X_H^{(k)}$ from $D_k$
5:      Form explored set $U_k \leftarrow X_H^{(k)} \cup X_M^{(k)}$
6:      Compute per-round normalisation statistics $(\mu_k, \sigma_k)$ over $U_k$ and define $\phi_k(\cdot)$
7:      **for all** anchors $x \in X_H^{(k)}$ **do**
8:          Find neighbour set $N_k(x)$ as the `Knn`-NN of $x$ in $U_k$ using $\|\phi_k(x) - \phi_k(\cdot)\|_2$
9:          Set radius $r_k(x)$ to the $q$-quantile of $\{\|\phi_k(x) - \phi_k(x')\|_2 : x' \in N_k(x)\}$
10:         Compute region difficulty $\tilde{d}_k(x)$ from demonstrated steps covered by the region
11:      **end for**
12:      Greedy-select $Q_k \subseteq X_H^{(k)}$ with $|Q_k| \leq L$ using the teaching gain in Sec. 4.2
13:      Render $Q_k$ as circles (centered at anchors, radius $r_k$, colour $\propto \tilde{d}_k$)
14:      Collect new demonstrations $\Delta D_k$ from the teacher and update $D_{k+1} \leftarrow D_k \cup \Delta D_k$
15:      Compute step weights $w_k(\cdot)$ from region difficulty (Sec. 4.3) and clip by $w_{\max}$
16:      Train $\pi_{\theta_{k+1}} \leftarrow A(D_{k+1})$ using weighted per-sample imitation losses
17: **end for**

---

strations and only the most recent rollout set used for the current guidance.

**Anchor candidates, neighbours, and radius.** Anchors are restricted to demonstrated states (*i.e.,* teacher-provided states), while neighbourhood queries are computed over the explored set. KNN search uses Euclidean distance in the per-round normalised observation space $\phi_k(\cdot)$ (defined in the main paper). We use `Knn = 30` neighbours and set each anchor radius to the $q=0.95$ quantile of its neighbour distances. This quantile choice avoids sensitivity to outliers while producing stable region sizes.

## A.3. Baselines

This section specifies implementation details for the baselines in Section 5. All baselines use the same learner, training budget, and round protocol as the main paper, and differ only in how new demonstrations are collected (interactive baselines) or how a fixed dataset is reweighted during training (offline curation). *No performance feedback* is provided

*Table 3.* Core diffusion policy and training hyperparameters.

| Component | Setting |
|---|---|
| Backbone | Transformer encoder–decoder |
| Encoder / decoder layers | 1 / 2 |
| Embedding dim / heads | 256 / 4 |
| Dropout (attn/resid) | 0.1 / 0.1 |
| Output head | Linear output |
| Time embedding | Sinusoidal + 2-layer MLP (Mish) |
| Weight init | Normal(0, 0.02) |
| Action sequence length | 20 |
| Chunk size / exec horizon | 20 / 20 |
| Action representation | Delta position; gripper abs. for pick-and-place |
| Diffusion steps | 100 |
| Beta schedule | Cosine ($s=0.008$), clipped to [0, 0.999] |
| Parameterisation | $\epsilon$-prediction |
| Denoised clipping | Enabled |
| Loss | L2 (MSE), weighted reduction |
| Optimizer | AdamW |
| Learning rate | $5 \times 10^{-4}$ |
| Betas / weight decay | (0.9, 0.95) / 0.0 |
| Batch size / epochs | 64 / 100 |
| Train/val split | 80/20 |
| LR schedule | CosineAnnealingLR |
| EMA | decay=0.995, update every step (used for inference) |
| Seed | 42 |

*Table 4.* Task-specific observation and action dimensions. Actions are delta-position (velocity) commands; pick-and-place additionally predicts a gripper command (absolute).

| Task | Obs dim | Act dim | Observation definition |
|---|---|---|---|
| Avoiding | 2 | 2 | $[\text{ee}_x,\ \text{ee}_y]$ |
| Pushing | 6 | 2 | $[\text{ee}_x, \text{ee}_y,\ \text{cube}_x, \text{cube}_y,\ \sin(\theta), \cos(\theta)]$ |
| Alignment | 6 | 2 | $[\text{ee}_x, \text{ee}_y,\ \text{cube}_x, \text{cube}_y,\ \sin(\theta), \cos(\theta)]$ |
| Pick-and-place | 8 | 4 | $[\text{ee}_{xyz},\ \text{grip},\ \text{cube}_{xyz},\ \theta]$ |

to teachers beyond the method-specific guidance.

**ASkDAgger-style uncertainty-gated querying.** We implement a selective querying baseline following the "query-when-uncertain" style of ASkDAgger. At each round, after training $\pi_{\theta_k}$, we roll out the learner to collect on-policy trajectories and record the visited states $\{x_t\}$. Each visited state is assigned an uncertainty score $u(x_t)$, and the top-4 states with highest uncertainty are selected. The environment is then reset to each selected snapshot state in turn, and the teacher provides a continuation demonstration from that state (*i.e.,* a complete expert trajectory segment from the snapshot until termination/success). These 4 continuation trajectories constitute the round's newly collected demonstrations.

*Uncertainty score.* Since our learner is a stochastic diffusion policy, we estimate uncertainty by repeated action sampling with different diffusion noise seeds at the same input state. Concretely, we sample $M$ first-step actions $\{a^{(m)}(x)\}_{m=1}^M$

from $\pi_{\theta_k}(x)$ and compute

$$u(x) \;=\; \mathrm{tr}\Big(\mathrm{Cov}\Big(\{a^{(m)}(x)\}_{m=1}^M\Big)\Big), \qquad (14)$$

*i.e.,* the trace of the empirical covariance of the sampled actions (equivalently, summed per-dimension variance). In practice, we score states at every environment step along the rollout and break ties deterministically (first occurrence).

**Easy-to-hard curriculum.** We implement an easy-to-hard baseline that stages demonstration collection over predefined task-specific difficulty bins. Early rounds allocate data collection to easier bins, then switch to the default task distribution in later rounds (Section 5). Difficulty bins are defined per task and visualised in Figure 8. Teachers always demonstrate from freshly sampled task instances within the current bin and otherwise follow the same interface and collection budget.

**Offline curation baselines (training-only).** To isolate the effect of training-time data selection, we evaluate two dataset reweighting heuristics on a fixed demonstration

*Table 5.* Episode horizons $H$ and workspace bounds per task.

| Task | Horizon $H$ | Workspace (m) |
|------|-------------|---------------|
| Avoiding | 400 | $x \in [0.2, 0.8],\ y \in [-0.5, 0.5]$ |
| Pushing | 400 | $x \in [0.2, 0.8],\ y \in [-0.4, 0.4]$ |
| Pick-and-place | 600 | $x \in [0.2, 0.8],\ y \in [-0.4, 0.4],\ z \in [0.03, 0.5]$ |
| Alignment | 1000 | $x \in [0.2, 0.8],\ y \in [-0.4, 0.4]$ |

dataset. The dataset itself is unchanged; only per-trajectory or per-sample weights applied in the loss differ.

*Trajectory-quality weighting.* Each trajectory $\tau$ is assigned a scalar quality score based on (i) outcome and (ii) efficiency. Let $\mathbb{1}\{\tau \text{ succeeds}\}$ indicate success and let $T(\tau)$ denote completion time in environment steps for successful episodes (undefined for failures). We compute a normalised efficiency term $\eta(\tau)$ (higher is better) and map it to a non-negative weight:

$$s(\tau) = \mathbb{1}\{\tau \text{ succeeds}\} + \lambda\, \eta(\tau), \qquad (15)$$

$$w(\tau) = \mathrm{clip}\big(1 + \beta\, s(\tau),\ 1,\ w_{\max}\big), \qquad (16)$$

where $\lambda$ controls the outcome–efficiency tradeoff, $\beta$ scales weighting strength, and $w_{\max}$ matches the global clipping used elsewhere. The per-sample imitation loss for steps within $\tau$ is multiplied by $w(\tau)$.

*Cluster-balanced reweighting.* We cluster demonstrations into behavioural groups and reweight samples inversely to cluster frequency. Each trajectory $\tau = \{z_t\}_{t=1}^{T}$ (where $z_t$ concatenates the recorded state-action vector at time $t$) is mapped to a fixed-length feature vector by (i) linearly re-sampling the sequence to $L{=}50$ timesteps and (ii) flattening the result:

$$f(\tau) = \mathrm{vec}(\mathrm{Resample}(\tau, L{=}50)) \in \mathbb{R}^{50D}. \qquad (17)$$

We standardise features across the dataset with z-score normalisation and compute a 2D PCA projection for clustering. We then apply $K$-means in the PCA space with $K{=}2$ clusters to obtain a cluster assignment $c(\tau) \in \{1, 2\}$. Let $n_c$ denote the number of trajectories in cluster $c$. Each trajectory receives weight

$$w(\tau) \propto \frac{1}{n_{c(\tau)}}, \qquad (18)$$

followed by normalisation to preserve unit mean weight and clipping to $w_{\max}$. The resulting trajectory weight is applied multiplicatively to the per-sample imitation loss for all steps in $\tau$.

**No-guidance baseline.** In the no-guidance condition, teachers provide demonstrations using their own judgement under the same collection budget and interface, but without any overlay or querying prompts.

## A.4. Evaluation Metrics and Protocol

This section specifies the exact evaluation rules used to compute success rate, normalised completion time, robustness under distribution shift, and behavioural diversity.

**Success criteria and horizons.** Table 6 summarises the task-specific success definitions and the episode horizons used for evaluation (and for normalising completion time). Episodes terminate upon success or when the horizon $H$ is reached.

**Normalised completion time.** For each successful episode, we record the first timestep $t_{\mathrm{succ}}$ at which the success condition becomes true, and report

$$t_{\mathrm{norm}} = \frac{t_{\mathrm{succ}}}{H}. \qquad (19)$$

Mean and standard deviation of $t_{\mathrm{norm}}$ are computed across successful test episodes; unsuccessful episodes are excluded from the completion-time statistic.

**Robustness under distribution shift (±10%).** To evaluate robustness, we perturb the initial state used for evaluation while keeping the trained policy fixed. For each test initialisation $x_0$, we sample a perturbed initial state

$$\tilde{x}_0 = x_0 + \delta, \qquad \delta \sim \mathrm{Uniform}(-0.1\,\Delta,\ 0.1\,\Delta), \quad (20)$$

where $\Delta$ denotes the per-dimension range of the corresponding variable in the default test initialisation distribution. Following Section 5, perturbations are applied to: (i) the end-effector start position for AVOIDING, (ii) the cube start position for PUSHING and PICK-AND-PLACE, and (iii) both cube position and orientation for ALIGNMENT. All methods use the same perturbation magnitude and the same perturbed test set.

**Behavioural diversity.** For AVOIDING and PUSHING, we quantify behavioural diversity by assigning each rollout to one of two known strategy modes using a fixed geometric rule (Figure 9 shows two strategies used in the simulation setting, Figure 10 shows the multiple strategies in the real robot setting), then computing the balance score (Figure 9). Concretely, we run 50 evaluation rollouts per policy and obtain counts $(n_1, n_2)$ for the two strategy labels. We report

$$r = \frac{\min(n_1, n_2)}{\max(n_1, n_2)} \in [0, 1], \qquad (21)$$

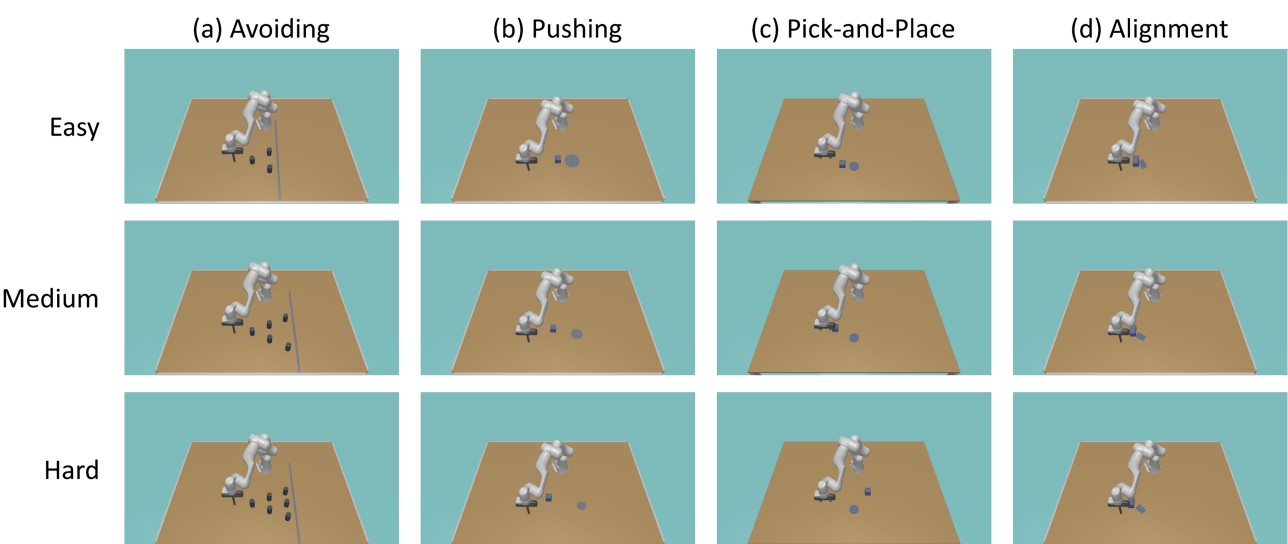

*Figure 8.* Curriculum Learning setting.

*Table 6.* Task-specific success definitions and evaluation horizons. Horizons match the experimental setting in Sec. 5.

| Task | Horizon $H$ | Tolerance(s) | Success rule (episode succeeds if) |
|---|---|---|---|
| Avoiding | 400 | goal radius $\leq 0.02$m | The end-effector reaches the goal region, *i.e.*, $\|p_{ee} - p_{goal}\|_2 \leq 0.02$. The episode is marked as failure if any collision occurs with obstacles during the rollout. |
| Pushing | 400 | position $\leq 0.02$m | The cube reaches the target region, *i.e.*, $\|p_{cube} - p_{target}\|_2 \leq 0.02$ at any time before timeout. |
| Pick-and-Place | 600 | position $\leq 0.02$m | The object is successfully placed at the target, *i.e.*, $\|p_{cube} - p_{target}\|_2 \leq 0.02$ with the cube resting in the placement region (no longer held by the gripper at the end of the episode). |
| Alignment | 1000 | position $\leq 0.02$m; angle $\leq 5°$ | The cube is aligned to the target pose: $\|p_{cube} - p_{target}\|_2 \leq 0.02$ and $|\theta - \theta^\star| \leq 5°$ (with $\theta^\star$ the task-specific target orientation). |

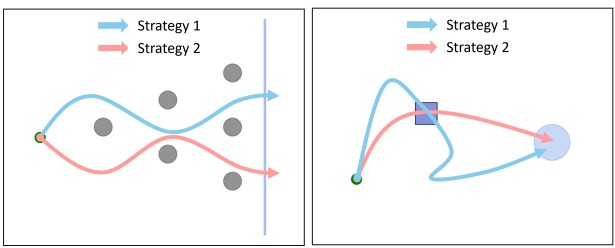

*Figure 9.* Strategy diversity in the simulation experiment.

where $r=1$ indicates perfectly balanced strategy usage and smaller values indicate mode collapse. For AVOIDING, the strategy label is determined by whether the end-effector passes the obstacle on the left or right side, computed by the sign of the lateral offset at the closest approach point. For PUSHING, the label is determined by whether the cube is approached from the left or top side relative to the target direction, computed from the sign of the cube's lateral displacement at first contact.

## B. Human Study and Interface Details

### B.1. Study Design and Participants

This appendix provides additional details for the user study protocol in Section 5. Ethical approval was obtained prior to the study, and approval documentation will be provided upon publication.

**Participants.** We recruited $N=30$ participants. Most participants were between 22 and 35 years old, with a small number of participants in ages 37, 39, and 55. The gender distribution was 8 female and 22 male participants. Only three participants reported prior exposure to LfD. The remaining participants came from other backgrounds and reported limited familiarity with LfD and diffusion policies. Table 7 summarises the participant demographics.

**Assignment and session structure.** Participants were randomly assigned to a condition with equal group sizes. Each session followed the same structure: (i) brief task instruc-

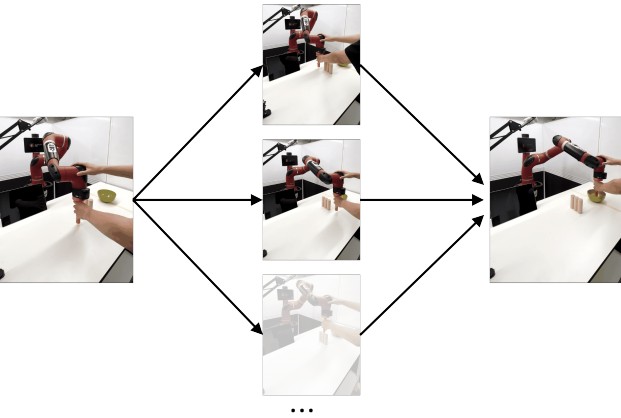

*Figure 10.* Strategy diversity in the real robot experiment.

*Table 7.* Participant demographics summary.

| Statistic | Value |
| --- | --- |
| Total participants | 30 |
| Age range | 22–55 |
| Primary age band | 22–35 (majority) |
| Female / Male | 8 / 22 |
| Prior LfD experience | 3 / 30 |

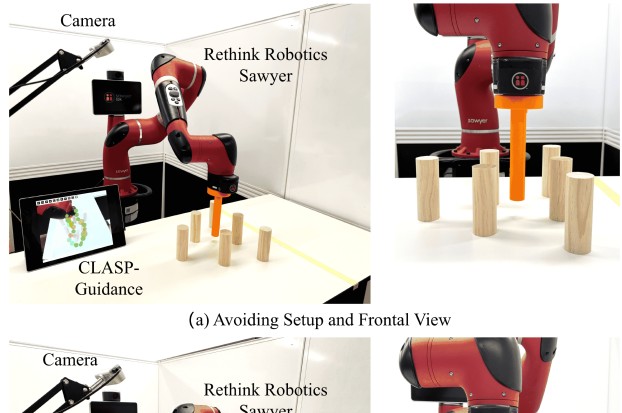

(a) Avoiding Setup and Frontal View

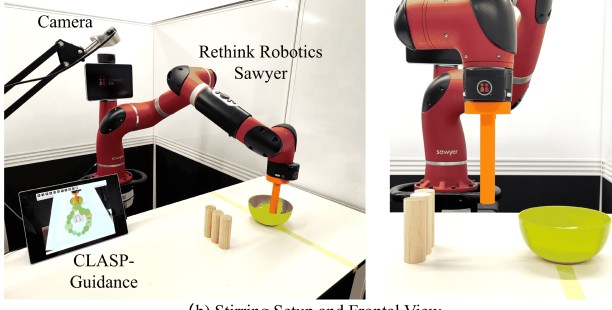

(b) Stirring Setup and Frontal View

*Figure 11.* Real-robot setting for AVOIDING and STIRRING tasks.

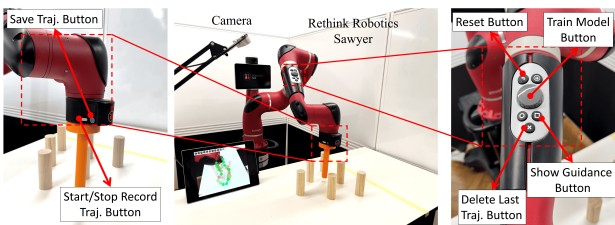

*Figure 12.* Control and interaction methods and buttons.

tions, (ii) optional short warm-up interaction to familiarise with controls (not counted toward the five rounds), and (iii) five teaching rounds.

### B.2. Teacher Interface and Instructions

**Interface overview.** Participants provide demonstrations through a shared teleoperation interface. The interface displays a live task view with the goal marker and relevant objects, as shown in Section 6's simulation and real-robot guidance view and Figure 11. In the CLASP-Full condition, guidance is rendered as an overlay of circular regions projected onto the workspace, where circle radius indicates local region extent and colour intensity indicates relative difficulty. In the No-Guidance condition, the same interface is used but the overlay is omitted.

**Guidance interaction.** Participants are free to demonstrate anywhere in the workspace in both conditions. The guidance overlay does not constrain the demonstrator, but highlights regions that are expected to be most informative. Participants are instructed that the overlay is intended to suggest *where* demonstrations may be most useful, but that demonstrations should remain task-valid.

**Round procedure.** Each round consists of a fixed number of demonstrations collected sequentially. A demonstration terminates on task success, timeout, or user reset. After each round, the learner is updated offline, and the next round begins with the same interface. **No scores, learning curves, or quantitative improvement signals are shown to participants during the session**.

### B.3. Logging and Quality Control

For each demonstration, we log the full state-action trajectory, episode outcome (success/failure), and completion time. For uncertainty-query baselines, we additionally log the selected snapshot states used for teacher continuation. We discard incomplete demonstrations caused by simulator resets or recording failures and immediately re-collect an episode under the same round budget. No participants were excluded from the analysis.

## C. Real-Robot Setup

### C.1. Hardware and Control Stack

All real-robot experiments are conducted on a Rethink Sawyer arm controlled through ROS Noetic. The end-effector is augmented with a lightweight 3D-printed rod to

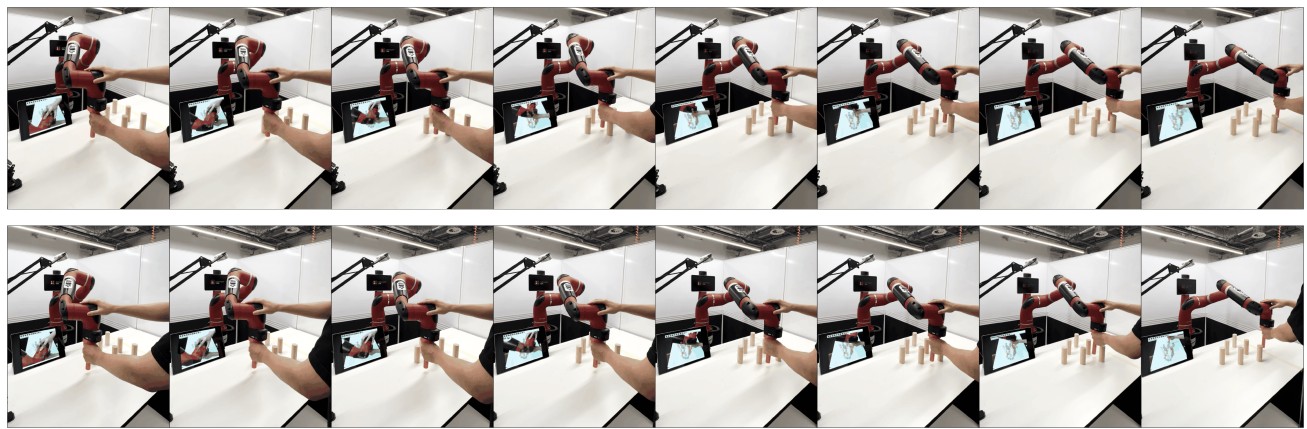

*Figure 13.* Kinesthetic demonstration with CLASP guidance.

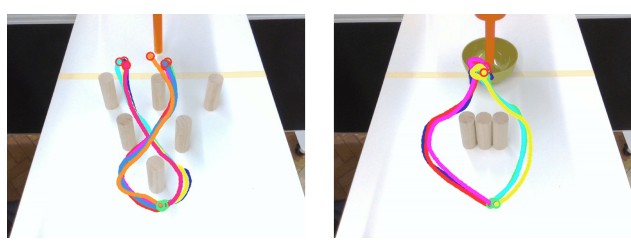

*Figure 14.* Real-robot demonstrations.

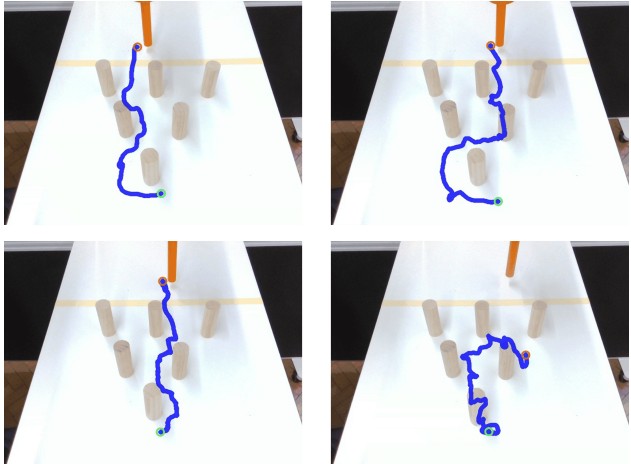

*Figure 15.* Real-robot learner rollouts.

start/stop recording, resetting the environment between trials, and triggering the policy update between rounds, as shown in Figure 12. This design avoids external controllers and provides a consistent interaction procedure across participants.

### C.2. Guidance Overlay Deployment

Figure 14 and Figure 15 show examples of the overlay view of demonstrations and learner rollouts. To deploy CLASP on the real robot, we draw the selected anchor regions directly on the live RealSense camera view shown to the teacher. We calibrate the RealSense once before the study using a standard checkerboard. This gives us a mapping from the robot's workspace coordinates to pixel coordinates in the camera image, and also accounts for a small fixed offset between the camera frame and the robot frame. Using this mapping, we project each anchor location and its radius into the image and render the corresponding guidance circles so that they align with the robot workspace.

increase reach and provide a consistent point of interaction with the workspace objects. A fixed Intel RealSense camera is used to capture a live top-down view of the working area, which is streamed in real time to a Microsoft Surface laptop that serves as the teacher display.

**Demonstration collection.** Demonstrations are kinesthetic: participants directly move the robot arm to complete the task while the system records the corresponding robot trajectories, as shown in Figure 13. The workflow is controlled using the physical buttons on the Sawyer robot, including

