# OpenReview forum: "Making Learner Weakness Actionable for Learning from Demonstration with Novice Teachers"
_ICML.cc/2026/Conference — ICML 2026 regular_

### Official Review · Reviewer_oXhc · 2026-02-23

**Soundness:** 3
**Presentation:** 2
**Significance:** 3
**Originality:** 3
**Overall Recommendation:** 4
**Confidence:** 3

**Summary:**

This paper proposes Collaborative Learning with Anchored State-space Partitions (CLASP), which is an interactive imitation learning method where the learner agent can request online demonstrations, but with a limited budget and a learner-agnostic "novice (human) teacher". The goal is to build a dataset minimizing per-step teaching risk within the demonstration budget. CLASP runs in iterations; for each iteration, a teaching representation is shown, which are spheres with human demonstration observations as center, locally adaptive KNN-based radius and soft maximum loss as difficulty. Based on the representation, two steps are conducted: CLASP-Teach, which illustrates the most representative regions for the human teacher to refer to; CLASP-Train, which is a weighted imitation based on difficulty of the related region. On several simulated and real robot experiments, the proposed method outperforms baselines.

**Compliance With Llm Reviewing Policy:**

Affirmed.

**Final Justification:**

I think the rebuttal fully addresses my concerns. Overall, I think the paper does a good work and is above the acceptance bar, but several potential concerns in the paper listed as the weaknesses prevents me from directly accepting this paper. With further explanation in the rebuttal, these concerns are addressed.

**Key Questions For Authors:**

Please refer to the weakness section for my other questions. I have one extra question: How is the region average difficulty for weight calculated at the first iteration, where there is no past anchor (line 251-253)?

**Limitations:**

Yes.

**Strengths And Weaknesses:**

**Strengths**

1. The problem that the paper tries to address is interesting and useful: active few-shot imitation learning from a non-expert human. This could be potentially useful for future embodied AI data labelers.

2. The empirical evaluation seems solid; it clearly shows the advantage of the proposed method both in simulated environment and real-life, with ample visualization (Fig. 3 / Fig. 5) that clearly shows how the method aids human teacher to better guide the robot agent. The detailed implementation details in the appendix makes the evaluation more reproducible.

**Weaknesses**

1. The CLASP-train loss seems to align the exact same time step of demonstration between the student's rollout and the demonstration (Eq. 13). This can be potentially limiting as it requires the same initialization. For example, consider a task where a robot need to run in a circle but the initial state is randomized on each rollout; in this case, $a_t$ should not be aligned with $x_t$, but with other states in the demonstration.

2. The definition of demonstration budget is inconsistent. In Eq. 1, the budget is defined by "the total number of labelled time steps" $E_T(D)=\sum_{\tau\sim D}|\tau|$ (line 151), and this is supposed to be no more than $B$. However, In the experimental setting (line 266-267), the budget is defined as $B\in\{4,8,12,16,20\}$ **trajectories**, with each trajectory's horizon spanning over 400-1000 steps (line 309-310).

3. If the participants never have any robotic teleoperation experience, the novice teacher may learn teleoperation during experiment; thus, their level of "teaching" expertise may vary with the order of the experiment, and affect final performance.

4. Fig. 5 shows that with CLASP, the teacher focuses more on the weaker strategy region - but there might be a possibility that the teacher should not always cater to the student's bad practice. For example, consider an object manipulation task where the agent needs to move a needle behind a wall with a small hole. Pushing the needle through the hole is a feasible solution, but is much slower and more difficult if one can simply carry the needle and bypass the wall from aside. Suppose the student now choose each strategy uniformly, with 10% success rate on pushing and 90% success rate on bypassing the wall. In this case, the teacher should not focus on teach the student how to push the needle through the hole, but should instead focus on letting the student bypass the wall.

**Minor Weaknesses**

1. Fig. 2 is a bit confusing: as the paper mentions "This shared 'map' supports two operations in each round: CLASP-Teach ... and CLASP-Train", and they are listed in parallel on Fig. 2, it seems as if one of the action is selected each round, but judging from Alg. 1 they are both conducted. I would suggest to modify the figure and/or the text to make it clear that they are both conducted for every round.

2. The "per-step loss" is not explained clearly in the paper. While some places indicate that it is probably L2-loss (e.g. line 556 and Tab. 3), this should be more clearly stated.

3. The running title on the top of each page is still "submission and formatting instructions for ICML 2026".

---

> ### Author Rebuttal · Authors · 2026-03-30
>
> We thank the reviewer for the careful reading and for recognizing the relevance of the problem setting and the strength of the empirical evaluation. We respond to each point below.
>
> **W1: Timestep alignment in CLASP-Train loss (Eq. 13).**
>
> We believe this concern comes from an ambiguity in how Eq. 13 is presented, and we are happy to clarify it. The loss does not align rollout timestep $t$ with demonstration timestep $t$. It is standard weighted behavioural cloning over demonstrated pairs.
>
> Every term involves a *demonstrated* pair $(x_t, a_t)$. The weight $w_k(x_t)$ comes from the difficulty of the nearest anchor region, which is defined independently of rollout timesteps. The separation is deliberate: rollouts inform *where the learner currently goes* (shaping region geometry via $U_k$ in Eq. 3), while the loss only involves *what the teacher demonstrated*. This keeps the difficulty signal grounded in teacher-relevant parts of the state space.
>
> To address the circular-task scenario directly: training evaluates the policy at each demonstrated state-action pair independently. No temporal correspondence between rollouts and demonstrations is needed.
>
> **W2: Budget definition inconsistency.**
>
> Thank you for pointing this out. We should have stated this more clearly. Eq. 1 uses timesteps for generality (since trajectories could have different lengths in other settings), while experiments report trajectory count $B$. Because our task horizons are fixed, $B$ trajectories is exactly $B \times H$ timesteps. We will state this mapping explicitly in the revision.
>
> **W3: Order effects from learning teleoperation.**
>
> This is a valid concern for any human-subject study. Our between-subject design means each participant sees exactly one condition, so skill improvement over rounds is not confounded with condition assignment. We also include a warm-up phase (Appendix B.1) so that participants are comfortable with the interface before data collection begins. Most importantly, the *No-guidance* group also completes five rounds of practice under the same protocol. The performance gap between CLASP and No-guidance at the same round number therefore reflects the effect of guidance, not accumulated practice.
>
> **W4: Should the teacher always focus on weak strategies?**
>
> We appreciate this point and agree it is an important aspect to discuss. CLASP's difficulty signal measures deviation from *demonstrated* behaviour, not task optimality, because our setting assumes no reward access and the teacher's demonstrations define what "correct" means. This constraint is inherent to reward-free imitation rather than a CLASP-specific choice.
>
> At the same time, our results suggest that under tight budgets, focusing on the underrepresented strategy actually helps. CLASP achieves the highest success rate (Figure 4) and the highest strategy diversity (Appendix Figure 9) simultaneously, which would not happen if targeting weaker strategies were harmful. We believe the reason is that under small budgets, the main bottleneck is skewed data coverage rather than strategy quality. When both strategies are reasonable (as in our benchmarks), targeting the underrepresented one fills exactly the gap that limits the policy. We thank the reviewer for the needle-through-hole example. In practice, CLASP only guides toward strategies the teacher has already chosen to demonstrate. If one approach is clearly inferior, the teacher is unlikely to show it in the first place, and CLASP would never surface it. Even when both strategies are demonstrated, the guidance is a suggestion, not a constraint. A teacher who notices a region staying red round after round can decide to stop investing in that approach. The extreme case where strategies differ vastly in quality is a different regime where reward signals would be needed, and we will discuss this boundary in the limitations section.
>
> **Q: First-iteration difficulty.**
>
> CLASP does not provide guidance when it has no evidence to work with, so in round 0 the teacher demonstrates freely without guidance, producing $D_0$. We train the first policy on $D_0$, and the demonstrated states become the first anchors. At that point we have both anchors and a trained policy, so difficulty is computed normally via LogMeanExp. The first round of guidance (round 1) is therefore already informed by the learner's actual mistakes on the initial data.
>
> **Minor issues.** We thank the reviewer for catching these and will fix Fig. 2, clarify the per-step loss as L2 (MSE), and correct the running title. We will also review notation consistency throughout to address the presentation concerns more broadly.

---

> > ### Author Rebuttal · Reviewer_oXhc · 2026-04-01
> >
> > I thank the authors for the detailed explantion. The rebuttal addresses my concerns, and I will increase my score from 3 to 4.

---

> > > ### Author Response · Authors · 2026-04-01
> > >
> > > We appreciate the recognition that the problem setting is **interesting and useful**, and that the empirical evaluation **clearly supports** the method. We are also grateful that the rebuttal addressed the reviewer’s concerns. Finally, we thank the reviewer for the careful review and thoughtful feedback.

---

### Official Review · Reviewer_pde9 · 2026-03-06

**Soundness:** 3
**Presentation:** 3
**Significance:** 3
**Originality:** 3
**Overall Recommendation:** 4
**Confidence:** 4

**Summary:**

To address the issue of limited demonstration budgets (constraints on time, rounds, and trajectory counts), the authors propose CLASP (Collaborative Learning with Anchored State-space Partitions). This framework “translates” the remaining weaknesses of learners into executable guidance for teachers. After each training round, CLASP constructs a region-level behavior map by combining (1) states appearing in the teacher's existing demonstrations and (2) states accessed by the learner's current policy rollout. Each region is anchored to a state from the teacher's demonstration, with its neighborhood defined by a locally adaptive radius. The “difficulty” score is aggregated from the learner's imitation error within that region, unifying “what has been covered” and “where learning is deficient” into a compact, visual representation.

Based on this map, CLASP further introduces two mutually coupled mechanisms:
- CLASP-Teach: Greedily selects a set of “difficult yet non-redundant” representative regions from overlapping areas to present to the teacher (visualized as circular highlights in the workspace), indicating where supplementary demonstrations in specific behavioral/state neighborhoods are most likely to enhance learning;
- CLASP-Train: During training, it applies difficulty-aware weighting to demonstration samples, enabling the learner to prioritize still-challenging regions within limited computational and data budgets. This aligns “what to teach” with “how to learn” using a single spatial map.

Experiments validated these mechanisms across multiple MuJoCo manipulation tasks and on Sawyer robots, using novice participants as teachers in turn-based budgeted teaching protocols.

**Compliance With Llm Reviewing Policy:**

Affirmed.

**Key Questions For Authors:**

1. Robustness evaluation scope. Distribution shift is implemented as ±10% perturbation of initial states. How sensitive are results to shift magnitude (e.g., ±20%) or different shift types (e.g., obstacle placement, friction/contact changes), especially for contact-rich tasks like ALIGNMENT?

2. Hardware experiments compare CLASP only to No-Guidance. Could you include at least one stronger baseline in hardware or justify why they are infeasible?

**Limitations:**

yes

**Strengths And Weaknesses:**

Strengths:
- In real deployments or crowd-sourced demonstrations, “limited budget + non-expert teachers” is common; novices often provide task-valid trajectories that are not maximally informative for the current learner. Taking into account the bottlenecks in reality.
- Translating learner failure modes into teacher-facing guidance (instead of only internal gating) could influence future interactive LfD UIs and teaching protocols.
- CLASP-Teach greedily selects representative regions via a “difficulty × marginal coverage” objective (human-interpretable visualization), while CLASP-Train derives sample weights from the same map to emphasize still-difficult regions during learning. Using one shared map reduces the mismatch between “what to teach” and “how to learn.”

Weaknesses:
- The “why it works” argument is mostly intuitive/motivational. The region map can be interpreted as approximating machine teaching via local difficulty plus coverage, but the paper provides limited property-based analysis (e.g., assumptions under which region selection approximates optimizing a proxy objective; or a more formal robustness discussion for softmax/quantile radius choices).
- CLASP-Teach’s greedy selection is heuristic; the relationship between greedy gain (e.g., $G_k$) and final imitation-risk reduction is not made explicit (e.g., via bounds or a surrogate improvement measure).
- The real robots were only compared against the No-Guidance baseline, not against stronger baselines. This makes the evidence for “sim-to-real also holds” insufficient.

---

> ### Author Rebuttal · Authors · 2026-03-30
>
> We appreciate the thorough review and the recognition that translating learner failures into teacher-facing guidance could influence future LfD interfaces. We agree the design choices deserve stronger grounding and address each point below.
>
> **W1: Limited property-based analysis for softmax and quantile choices.**
>
> We thank the reviewer for pushing on this. We should have made the statistical motivation more explicit in the paper, and we clarify here. For region difficulty: $\tilde{d}_k(x)$ is the log-moment generating function of the per-step loss distribution within the region. Via the Donsker-Varadhan variational representation, this equals the worst-case expected loss under KL perturbation of the empirical distribution. In practice, this means regions with high loss variance are treated as difficult: the exponential upweights the tail without the instability of a hard maximum. It sits between mean and max, with the gap driven by within-region loss variance.
>
> For the quantile radius: the $q{=}0.95$ quantile of KNN distances provides a stable middle ground between the mean (insensitive to outliers) and the maximum (too sensitive to a single one).
>
> The built-in ablation between CLASP-Full and CLASP-Teach confirms these choices matter in practice: both use identical demonstrations, and the only difference is whether difficulty-aware weights are applied. On easier tasks the gap is small, but on Alignment it reaches 0.10 at $B{=}20$ and grows to 0.12 under distribution shift (Table 1), which fits the robust risk interpretation.
>
> We also ran an ablation comparing four weight strategies (CLASP Region, Per-Step Loss, Uniform, Inverse-Loss) on the same CLASP-collected datasets at $B{=}20$ (full table in our response to Reviewer Qt4Y). Region-level weights are the best strategy on three of four tasks, with the largest gain on Alignment (0.66 vs. 0.58 for the next-best Per-Step Loss).
>
> **W2: Greedy selection lacks theoretical justification.**
>
> We appreciate this question, as it gives us a chance to clarify the formal picture. When difficulty is uniform across regions, CLASP-Teach's greedy selection reduces exactly to the classical maximum coverage problem, which has a $(1{-}1/e)$ approximation guarantee (Nemhauser et al., 1978). With non-uniform difficulty, the coverage structure remains submodular, but the difficulty multiplier makes the overall gain depend on selection order, so the standard guarantee does not directly apply. In practice, the difficulty ratio across active guidance regions is typically 2-5$\times$ (visible in Figure 5 headers), so the deviation from uniform is moderate.
>
> For downstream risk: demonstrations from CLASP-Teach's greedy selection consistently outperform those without guidance across all tasks (Figure 4), suggesting the selection targets the regions that matter most for policy improvement, even without a formal bound.
>
> **W3/Q2: Hardware baselines.**
>
> We believe that the simulation already covers all baselines and shows consistent CLASP advantages, so the hardware experiments were designed to test something specific: whether novice teachers actually respond to the camera-overlay guidance during kinesthetic teaching, and whether the performance trends hold on a physical robot. Table 2 shows they do. Adding another condition would require recruiting 5 new participants for full sessions on a physical Sawyer, with calibration and safety checks. We are exploring this and will include any available data in the revision.
>
> **Q1: Robustness to larger distribution shifts.**
>
> Following the reviewer's suggestion, we re-evaluated the trained policies at $\pm20\%$ perturbation (same protocol as Table 1):
>
> | Task | CLASP-Full | CLASP-Teach | No Guidance |
> |------|-----------|-------------|-------------|
> | Avoiding | 0.84 | 0.80 | 0.44 |
> | Pushing | 0.60 | 0.54 | 0.18 |
> | Pick-and-Place | 0.66 | 0.58 | 0.30 |
> | Alignment | 0.40 | 0.24 | 0.06 |
>
> The methods drop as expected with doubled perturbation, but CLASP's advantage does not shrink. On Alignment, which the reviewer specifically asks about, CLASP-Full maintains a clear lead while No Guidance nearly fails entirely. The Full vs. Teach gap also widens compared to ±10% (Table 1), consistent with the robust risk interpretation from W1: difficulty-aware weighting becomes more valuable as conditions get harder.

---

> > ### Author Rebuttal · Reviewer_pde9 · 2026-04-01
> >
> > I am very grateful to the authors for answering my questions.

---

> > > ### Author Response · Authors · 2026-04-01
> > >
> > > Thank you again for your careful review and constructive feedback throughout the process. We also appreciate the recognition that our work addresses a common and realistic setting, and that translating learner failures into teacher-facing guidance could help inform future interactive LfD interfaces and teaching protocols. We are grateful that our rebuttal and additional results addressed the reviewer’s concerns, and we appreciate the reviewer’s reconsideration in light of these clarifications. We will make sure these points are clearly incorporated in the final revision.

---

### Official Review · Reviewer_YRsz · 2026-03-13

**Soundness:** 2
**Presentation:** 3
**Significance:** 2
**Originality:** 2
**Overall Recommendation:** 3
**Confidence:** 3

**Summary:**

This paper proposed a guidance framework CLASP that helps teachers resolve imitation failures under a fixed demostration budget. Through building a compact map whcih exposes failure patterns and highlights where additional demonstrations are most likely to improve learing ,CLASP increases success by up to 20% over interactive and offline baselinses under the same demonstration budget , and improves robust success under distribution shift by 14-20% while mainitaining high behavioal diversity

**Compliance With Llm Reviewing Policy:**

Affirmed.

**Final Justification:**

Given the rebuttal, I keep my score unchanged

**Key Questions For Authors:**

See Weaknesses

**Limitations:**

Yes

**Strengths And Weaknesses:**

Strenghs:

1)The paper focuses on a key challenge in LfD—how to improve learning efficiency under limited demonstrations—and proposes a framework to guide teachers toward regions where the policy struggles.

2)By providing a region-level interface that summarizes coverage and difficulty, the framework clearly identifies what should be demonstrated next, significantly improving learning efficiency.

3)The design of CLASP-tech and CLASP-train effectively couples data collection with optimization.

4)The paper is well written and orgnized.

Weaknesses:

1)Assuming that demonstrations and rollouts tend to cluster around certain modes while leaving others sparse, it is unclear whether the learned regions accurately represent local portions of real behavior under limited demonstrations.

2)How to guarantee the  regions produced under KNN search is consistant with the demonstrations. Would the fixed KNN value influence the results?

3)While the current experiments are relatively simple, it would be important to validate the proposed framework in a wider range of task scenarios, particularly high-dimensional, multimodal, and expensive-to-acquire environments, to better demonstrate its generality.

---

> ### Author Rebuttal · Authors · 2026-03-30
>
> We appreciate the careful reading and the recognition that the Teach-Train coupling works effectively. We address each below with both design explanations and new sensitivity experiments.
>
> **W1: Do regions accurately represent local behaviour under limited demonstrations?**
>
> We would like to clarify how this works in practice, because the regions are built from considerably more data than the demonstrations alone. The explored set $U_k = X_H \cup X_M$ combines demonstration waypoints with the learner's own rollout states, so the KNN computation has more to work with than the raw demonstration count might suggest. Even a weak policy's rollouts are informative here, because they reveal where the learner actually goes, and the gap between rollout states and demonstrations is exactly what defines difficulty. This pool grows every round as new demonstrations and rollouts come in.
>
> The locally adaptive radius reinforces this. It uses the $q{=}0.95$ quantile of KNN distances, so regions automatically expand in sparse areas and shrink in dense ones. When demonstrations cluster around certain modes, regions in sparser areas grow wider rather than missing them. The CLASP-Full vs. CLASP-Teach comparison provides direct evidence that these regions capture something meaningful: on Alignment at $B{=}20$, applying region-based weights to the same demonstrations improves success from 0.56 to 0.66, which would not happen if the regions were uninformative.
>
> **W2: KNN consistency and sensitivity to K.**
>
> For consistency: the KNN search operates in a per-round normalised space $\phi_k$ (zero mean, unit variance over $U_k$), which makes neighbour relationships scale-invariant across tasks and rounds. Anchors are restricted to demonstrated states ($X_H$), so every region is centred on something the teacher actually showed, while the neighbourhood draws from the full explored set. This keeps regions grounded in teacher intent while reflecting what the learner has explored.
>
> For sensitivity: we kept the same demonstrations (collected under the default $K{=}30$ guidance) and re-computed region weights with $K_\text{nn} \in \{10, 20, 30, 50\}$ at $B{=}20$. This tests the training side specifically. We cannot vary $K$ for guidance without re-collecting data, but the guidance circles are anchored at demonstrated states regardless of $K$ and varying $K$ only adjusts the radius, so we expect the guidance-side effect to be limited.
>
> | Task | K=10 | K=20 | K=30 (default) | K=50 |
> |------|------|------|----------------|------|
> | Avoiding | 0.92 | 0.94 | 0.94 | 0.93 |
> | Pushing | 0.73 | 0.80 | 0.81 | 0.78 |
> | Pick-and-Place | 0.76 | 0.82 | 0.84 | 0.84 |
> | Alignment | 0.53 | 0.62 | 0.66 | 0.60 |
>
> Simpler tasks like Avoiding are insensitive because 20 neighbours already captures the region structure. Harder tasks show more spread: on Alignment, $K{=}10$ produces noisy regions that hurt training weights, while $K{=}50$ over-smooths and merges distinct failure modes. The quantile-based radius provides stability regardless of $K$, since it adapts to whatever neighbour set it receives.
>
> **W3: Task complexity and generality.**
>
> Our benchmarks involve multi-strategy behaviour (Appendix Figure 9), span 2D to 8D observations with 400 to 1,000-step horizons, and include contact-rich manipulation requiring $\leq$0.02m / $\leq$5$^\circ$ accuracy (standard D3IL benchmarks). We also validate on a physical Sawyer robot with kinesthetic teaching (Table 2, Figure 7), and CLASP's advantages hold under distribution shift (Table 1).
>
> High-dimensional observations remain the main open direction. CLASP operates on an embedding space $\phi_k$, so extending to images would involve replacing the feature extractor with a pretrained visual encoder. We think this is a well-scoped step, though KNN behaviour in higher-dimensional embeddings would need careful validation. We appreciate the reviewer highlighting this as a direction worth pursuing.

---

> > ### Author Rebuttal · Reviewer_YRsz · 2026-04-03
> >
> > Thank you for the clarification. The responses have partially addressed my concerns. I will keep my score unchanged.

---

> > > ### Author Response · Authors · 2026-04-03
> > >
> > > We thank the reviewer again for the thoughtful feedback and follow-up. We would greatly appreciate any indication of which specific point would benefit from further clarification, so that we can address them more clearly in the revision. If useful, we would be happy to provide additional details.

---

### Official Review · Reviewer_Qt4Y · 2026-03-13

**Soundness:** 3
**Presentation:** 3
**Significance:** 3
**Originality:** 3
**Overall Recommendation:** 4
**Confidence:** 4

**Summary:**

This paper formulates a new problem in online imitation learning: the learner has the budget to interact with the expert. To resolve this, the authors proposed a method named CLASP (Collaborative Learning with Anchored State-space Partitions). The key idea is to 1. Quantify the uncertainty via imitation loss. 2. Build region maps with KNN. 3. CLASP-Teach visualizes the regions for novice teachers to provide new demos. 4. CLASP-Train did difficulty-aware training. Results are tested on 4 simulated robot tasks along with a real robot setup.

**Compliance With Llm Reviewing Policy:**

Affirmed.

**Final Justification:**

Most of concern reslolved in rebuttal. Fig. 1 is much clearer than before. Thanks for the revision. Scores updated accordingly.

**Key Questions For Authors:**

1. Why the proposed weight a better choice than others? E.g., the imitation loss itself.
2. What should the novice teacher do to follow the visualization? Can you show if there are any uncertainty change after each interaction?

**Limitations:**

yes

**Strengths And Weaknesses:**

**Soundness**: The proposed problem and method are reasonable, no theoretical analysis, experimental results include both sim and real, with appropriate ablation. The author also claims their limitation.

**Presentation**: The structure of writing is overall easy to follow, but Fig1 is not that intuitive to understand what this paper is trying to resolve.

**Significance**: The problem is useful for human-robot interaction and data collection with human in the loop.

**Originality**: The method is mainly composed of two parts, CLASP-Teach and CLASP-Train. CLASP-Teach is a visualization method based on the KNN region split method; CLASP-Train is weighted imitation learning with weights computed from the region uncertainty. Weighted imitation learning is not new and the author does not well-claimed why their weight stragtegy is a better choce. (See problem)

---

> ### Author Rebuttal · Authors · 2026-03-30
>
> We thank the reviewer for the thoughtful feedback. We would like to clarify how we see the contribution of CLASP. Our claim is not that weighted imitation learning or KNN-based regions are individually novel. Rather, the key idea is to use a single shared region map to connect *teacher guidance* and *training emphasis*. This coupling is what allows the same representation to shape both what the novice teacher demonstrates and what the learner prioritizes during training. The human study is especially important here, because it shows that this shared representation changes how novices allocate demonstrations across rounds.
>
> **Q1: Why is the proposed weight a better choice than others (e.g., the imitation loss itself)?**
>
> We think the most direct evidence is the CLASP-Full vs. CLASP-Teach comparison. Both use the exact same guided demonstrations. The only difference is that CLASP-Full applies difficulty-aware weights during training. On easier tasks the gap is small (Avoiding: 0.94 vs. 0.92), but on harder tasks it opens up considerably (Alignment: 0.66 vs. 0.56 at $B{=}20$). Under distribution shift the difference grows further to 0.54 vs. 0.42 (Table 1), which fits the expectation that weighting matters most where the learner has more room to struggle.
>
> The paper also includes two other weighted BC approaches that help put this in context. Quality Weighting assigns trajectory-level weights based on task outcome and efficiency, and Cluster Weighting rebalances by inverse cluster frequency. Both are outperformed by CLASP-Full across all tasks (Table 1). What we find particularly telling is that even CLASP-Teach, which uses no weighting at all, beats both of them (Alignment under shift: 0.42 vs. 0.20 and 0.18). **Better data collection matters more than weighting unguided data, and region-level weights add further gains on top**.
>
> To test this directly, we ran an ablation comparing four weight strategies on the same CLASP-collected datasets at $B{=}20$:
>
> | Task | CLASP Region | Per-Step Loss | Uniform | Inverse-Loss |
> |------|-------------|---------------|---------|--------------|
> | Avoiding | 0.94 | 0.92 | 0.92 | 0.83 |
> | Pushing | 0.81 | 0.79 | 0.78 | 0.70 |
> | Pick-and-Place | 0.84 | 0.81 | 0.85 | 0.69 |
> | Alignment | 0.66 | 0.58 | 0.56 | 0.47 |
>
> Region-level weights outperform per-step loss on all four tasks, with the largest gap on Alignment (0.66 vs. 0.58) where per-step noise is most severe. Inverse-loss consistently hurts, confirming that the direction of emphasis matters. We believe the reason comes down to stability under small data: with $\leq$20 trajectories, individual step losses are noisy, and pooling over a KNN neighbourhood averages that out. LogMeanExp has a well-known connection to worst-case risk under distributional shift (Donsker-Varadhan), and the normalisation $w_k(x_t) = 1 + \tilde{d}_k(\alpha_k(x_t))/(\bar{d}_k + \varepsilon)$ adjusts itself as the policy improves, so weights do not need manual rescaling between rounds.
>
> **Q2: What should the novice teacher do? Can you show uncertainty change after each interaction?**
>
> The guidance shows up as circles overlaid on the workspace (Figures 3, 7), coloured by difficulty (red = hard, green = easy) and sized by coverage. Teachers are told to focus demonstrations near these challenging regions, but they can demonstrate wherever they think is right. It is a suggestion, not a constraint.
>
> Figure 5 shows how difficulty changes across rounds. The red and green boxes in each subplot report the average difficulty of the top-10 guidance regions for each strategy. As the teacher provides demonstrations in the targeted areas, difficulty there goes down. At the same time, CLASP surfaces the next most-difficult regions, and the teacher's allocation shifts accordingly. In other words, novices understand the visualisation well enough to act on it, and the underlying difficulty estimates update meaningfully as new data comes in.
>
> **Fig 1 clarity.** We apologise that Figure 1 does not convey this well enough. We will revise it to show the key distinction: existing interactive LfD methods decide *when* to query the teacher, while CLASP adds *what* to demonstrate and *how* to weight the result.

---

> > ### Author Rebuttal · Reviewer_Qt4Y · 2026-04-04
> >
> > The additional experiment and answer resolved my concern. Please consider including all these discussions in the revision. If you can further show me the revised version of Fig. 1 in an anonymous link, I can improve my score.

---

> > > ### Author Response · Authors · 2026-04-04
> > >
> > > Thank you again for your thoughtful feedback. For Fig. 1, our original intention was to provide only a very high-level comparison, but in doing so, the figure did not communicate the core gap clearly enough: prior interactive LfD can indicate **when** to query, but it does not provide novice teachers with actionable guidance on **what** to demonstrate next.
> > >
> > > Following your helpful suggestion, we prepared a revised anonymous version of Fig. 1 here: https://anonymous.4open.science/r/anonymousrepo-EF35/figure1.png. We will use this revised version in the final paper.
> > >
> > > If there is anything further that would make the figure clearer, we would be very grateful for the suggestion.

---

### Decision · Program_Chairs · 2026-04-30

**Decision:**

Accept (regular)

**Comment:**

This paper proposes a framework for interactive imitation learning that addresses learning under a fixed budget. The key contribution is a shared region map — built from teacher demonstrations and learner rollouts via locally adaptive KNN — that guides teachers on what to demonstrate next and emphasizes difficult regions during training. Empirical results span multiple MuJoCo manipulation tasks in the D3IL benchmark and a real Sawyer robot, showing improvements in success rate and gains in robustness under distribution shift.

The paper received three weak accepts and one weak reject. The reviewers broadly agree on the following positives: the problem is practically motivated and realistic, the shared map is a clean design choice, the empirical evaluation is solid with meaningful ablations, and the real-robot human study provides good external validity. Three reviewers were satisfied with the rebuttal and either maintained or upgraded their weak accept scores.

The primary remaining concerns to me is the scalability to harder settings — the benchmarks are relatively low-dimensional and the extension to image-based or multimodal tasks is only acknowledged as future work. Also the hardware experiments compare only against a no-guidance baseline. I think these concerns keep this paper at a weak accept rather than an accept, since imitation learning has been scaled up to much harder domains at this point.